# Pleiotropic Mucosal Innate Immune Memory in the Gastrointestinal Tract

**DOI:** 10.3390/ijms262010093

**Published:** 2025-10-16

**Authors:** Rachel H. Cohen, Sean P. Colgan, Ian M. Cartwright

**Affiliations:** 1Mucosal Inflammation Program, Department of Medicine, University of Colorado Anschutz Medical Campus, Aurora, CO 80045, USA; rachel.h.cohen@cuanschutz.edu (R.H.C.); sean.colgan@cuanschutz.edu (S.P.C.); 2Medical Scientist Training Program, University of Colorado Anschutz Medical Campus, Aurora, CO 80045, USA; 3Department of Medicine, VA Eastern Colorado Health Care System, Aurora, CO 80045, USA

**Keywords:** innate immune memory, mucosal inflammation, inflammatory bowel disease

## Abstract

Research in the past fifteen years has established that innate immune cells can develop immune memory, termed trained immunity. Trained innate immune cells exhibit distinct lasting epigenetic and metabolic changes that prime these cells upon repeated exposure. The gastrointestinal tract provides an important immunological barrier and is home to many innate immune cells, where trained immunity serves an essential role. This review summarizes what is currently known about the basic mechanisms behind innate immune memory, the roles of innate immune cells within the intestine, intestinal-specific trained immunity, and therapeutic potential for targeting trained immunity in the context of gastrointestinal disorders.

## 1. Introduction

The innate immune system has long been considered a rapid yet non-specific line of defense. This is in contrast to the highly specific and long-lasting memory responses of the adaptive immune system [1,2]. Emerging research has challenged this view by demonstrating that certain innate immune cells can exhibit memory-like characteristics known as trained immunity [3]. During trained immunity cells of the innate immune system, such as monocytes, macrophages, and neutrophils, undergo epigenetic and metabolic reprogramming following an initial stimulus and display an augmented response upon subsequent exposure [4,5].

Key features that distinguish trained immunity from adaptive immune memory are shown in Figure 1 to include antigen non-specificity, persistent epigenetic modulation (e.g., histone modifications, altered DNA methylation profiles), and metabolic shifts (e.g., increased glycolysis and glutaminolysis) [6,7,8,9,10]. Importantly, these changes can result in two distinct outcomes—either heightened reactivity (“training”) or suppressed responsiveness (“tolerance”), depending on the nature, intensity, and duration of the initial stimulus. Canonical training agents such as β-glucan and Bacillus Calmette–Guérin (BCG) vaccine promote proinflammatory and antimicrobial programs through mTOR- and HIF-1α-dependent metabolic remodeling [6,11,12,13,14], whereas repetitive or high-dose lipopolysaccharide (LPS) stimulation induces tolerance through itaconate accumulation and histone deacetylase activity [15,16]. Interestingly, trained immunity can promote enhanced effector responses, or tolerance, depending on the type of training and training context (Figure 1).

Recently, trained immunity has been separated into two distinct subsets, central and peripheral trained immunity. Central trained immunity refers to epigenetic and metabolic changes in hematopoietic stem cell progenitors in the bone marrow and their progeny, multipotent progenitors [11,17,18]. This training in the bone marrow allows for long-term innate immune memory. In contrast, peripheral trained immunity refers to changes in immune memory in tissues outside of the bone marrow, such as innate immune cells circulating in the gastrointestinal (GI) tract [19,20]. These distinctions are essential, as bone-marrow “central” training can influence systemic immune tone, whereas “peripheral” training at mucosal sites often reflects transient but functionally important adaptations shaped by local microbiota-derived signals Understanding the basic mechanisms of trained immunity is essential for harnessing innate immune memory in health and disease.

Mucosal surfaces, particularly in the GI tract, form one of the largest and most complex immune environments in the body [8,21,22]. The GI tract is constantly exposed to a multitude of antigens—both exogenous (dietary, environmental) and endogenous (microbial). The GI mucosal immune system must discriminate harmless or beneficial stimuli from potentially dangerous pathogens, all while performing its normal physiologic functions (nutrient extraction, water absorption, motility, etc.). The GI tract harbors an estimated 10^13^–10^14^ microbes, forming a major immunological interface [8]. Excessive or inappropriate inflammation can disrupt the gut barrier, leading to tissue damage or chronic inflammatory diseases [23,24,25]. Conversely, an underactive immune response risks uncontrolled pathogen invasion [22,26]. Because the gut is highly vascularized and drains into systemic circulation via the portal vein, local immune events can have systemic impacts, influencing metabolism, autoimmunity, and overall health [27].

The term “pleiotropy” in immunology reflects how a single immunological process can affect multiple pathways and cell types. In the context of trained immunity, epigenetic changes often upregulate genes that modulate a wide range of cellular functions—ranging from cytokine secretion and phagocytosis to metabolism and tissue repair [7,10]. In the GI tract, such pleiotropic immune function can be beneficial in combating a wide variety of infections, but may also heighten the risk of inflammatory disorders if misdirected or chronically activated [21,28]. These pleiotropic responses underscore the dual nature of trained immunity—protective in acute settings but potentially pathogenic when sustained, as seen in chronic colitis and other inflammatory conditions [29].

In this review, we explore the distinctive immunological landscape of the GI mucosal environment, the mechanistic underpinnings of trained immunity focusing on epigenetic and metabolic reprogramming, the benefits and risks (pleiotropic effects) of trained immunity in the gut, and potential therapeutic and clinical implications, including targeted manipulation of the microbiome and development of trained immunity–based interventions. Special emphasis is placed on distinguishing trained enhancement from tolerance and introducing the emerging concept of “trained tolerance,” wherein inflammation is restrained while epithelial repair programs remain epigenetically poised [10,15,30].

## 2. Mechanisms Underlying Innate Immune Memory (Trained Immunity)

In the early conceptualization of immune memory, only B and T lymphocytes were considered capable of “remembering” past antigen exposure. The discovery that innate immune cells (particularly monocytes, macrophages, and NK cells) can also display enhanced responsiveness challenged this dogma [1,2,4,5,10,11,19,20]. Trained immunity is induced by primary stimuli, commonly pathogen-associated molecular patterns or certain vaccines, that elicit a reprogramming event, rendering cells hyper-responsive to subsequent challenges. Notably, this reprogramming can last from days to weeks in circulating monocytes and potentially longer in tissue-resident macrophages [5,9,11,29].

Epigenetic and metabolic reprogramming lie at the core of trained immunity. Epigenetic changes include modifications to histone proteins (e.g., increased H3K4me3, decreased H3K9me3) and alterations in DNA methylation patterns, which regulate the accessibility of gene promoters and enhancers [2,5,7,9,16,31,32,33,34,35]. This dynamic chromatin remodeling is crucial for shaping the transcriptional responses of innate immune cells during both the induction and maintenance of their trained state. Recent work has underscored that histone deacetylases (HDACs) and other epigenetic regulators are key mediators in this process, fine-tuning the amplitude and duration of inflammatory responses as part of trained immunity [7,16,31,36]. Recent single-cell chromatin accessibility and multiomic studies confirm that these histone and methylation changes persist beyond the resolution of primary infection, maintaining an epigenetic ‘memory’ signature in human monocytes and intestinal macrophages [32,34,35].

On the metabolic side, shifts toward increased glycolysis, glutaminolysis, and alterations in the tricarboxylic acid (TCA) cycle support rapid ATP production for cytokine synthesis and effector functions [6,12,15,16,37] (Figure 1). These metabolic shifts often intersect with epigenetic pathways. For instance, intermediates of the TCA cycle can serve as cofactors (e.g., α-ketoglutarate) or inhibitors (e.g., fumarate) of histone- and DNA-modifying enzymes, thereby reinforcing a “poised” transcriptional state that can be recalled upon re-challenge. Through these interconnected epigenetic and metabolic circuits, trained innate cells such as macrophages, monocytes, and even neutrophils become primed to mount a more robust and rapid response when faced with a second encounter of the same or related stimulus. β-glucan and BCG training enhance glycolysis and mevalonate metabolism through mTOR and HIF-1α activation, leading to fumarate accumulation and H3K4me3 enrichment, whereas prolonged or high-dose LPS exposure drives tolerance through itaconate production, SIRT1 activation, and histone deacetylation [15,16].

Innate immune memory is strongly influenced by pattern recognition receptors (PRRs). Toll-like receptors (TLRs) on macrophages, DCs, and epithelial cells recognize microbial ligands such as LPS, peptidoglycan, and flagellin, activating transcription factors (e.g., NF-κB) that drive proinflammatory gene expression [26,37]. NOD-like receptors detect cytoplasmic microbial components and stress signals, and activation of the NLRP3 inflammasome triggers IL-1β release, a cytokine often upregulated in trained innate cells [26,37,38,39]. Concurrent stimulation of multiple PRRs can synergistically enhance or modulate trained responses, contributing to the pleiotropic nature of innate immune memory [26,37]. At mucosal surfaces, microbial metabolites such as short-chain fatty acids (SCFAs), indole derivatives, and secondary bile acids act through AHR and FXR signaling to bias cells toward trained or tolerant states, thereby linking microbial metabolism to the trained immunity [40,41,42,43,44,45,46].

## 3. Emerging Roles for Innate Immune Cells in Trained Immunity

Trained immunity encompasses a broad range of innate cell types that acquire long-lasting or context-dependent memory following exposure to microbial or inflammatory stimuli. While much of the early research emphasized monocytes and macrophages, it is now clear that neutrophils, dendritic cells, NK cells, and innate lymphoid cells (ILCs) can also undergo forms of functional reprogramming that shape subsequent immune responses and disease susceptibility. The following subsections highlight cell-specific mechanisms and their relevance to mucosal immunity.

### 3.1. Neutrophils and Trained Granulopoiesis

Although most early studies on trained immunity focused on monocytes, macrophages, and NK cells, neutrophils are increasingly recognized as exhibiting memory-like or “trained” phenotypes under select conditions [20,47,48]. Neutrophils have traditionally been regarded as short-lived responders with limited transcriptional flexibility, but newer findings suggest that exposure to specific stimuli—such as fungal β-glucans, LPS, or even certain vaccines (e.g., Bacille Calmette-Guérin (BCG))—can transiently alter neutrophil function during their relatively short lifespan [12,47,48,49]. Despite their short half-life, bone-marrow progenitors that give rise to neutrophils can themselves be epigenetically reprogrammed by systemic inflammatory cues such as IL-1β and GM-CSF—a process now termed “trained granulopoiesis.” This mechanism allows waves of newly generated neutrophils to inherit altered transcriptional and metabolic states long after the initial stimulus [12,20,50,51].

These alterations may include:

Enhanced Effector Responses: Priming for increased oxidative burst, degranulation, or production of neutrophil extracellular traps (NETs).

Epigenetic Changes: Although not as extensively characterized as in monocytes or macrophages, recent research indicates that neutrophils can undergo chromatin modifications that influence gene expression.

Prolonged Survival or Altered Migration: Under inflammatory or training conditions, some neutrophils may display prolonged survival cues and unique trafficking patterns within tissues, contributing to sustained immune alertness.

Such reprogrammed neutrophils can mount a more potent response if re-exposed to the same or related stimulus. The precise duration and mechanistic depth of this “trained” state in neutrophils remain active areas of research, but it is increasingly clear that neutrophils—like other innate immune cells—can be instructed by prior exposures in ways that enhance or skew subsequent inflammatory reactions [20,47,48,50,52]. Transcriptomic and epigenomic studies suggest that trained neutrophils exhibit enhanced chromatin accessibility at loci regulating oxidative metabolism, cytokine production, and NET formation, coupled with increased glycolytic flux and fumarate accumulation [12,48,50,53]. However, these reprogrammed states are short-lived and depend on continuous replenishment from reprogrammed progenitors [20].

In neutrophils, epigenetic remodeling is less thoroughly mapped than in monocytes or macrophages; however, short-lived changes in transcription factor binding and histone modifications have been described [20,48]. Stimuli like LPS can induce rapid transcriptomic adaptations, and select metabolic shifts—particularly involving glycolysis—that may underlie primed effector functions (e.g., increased production of reactive oxygen species). Likewise, the heme-containing enzyme myeloperoxidase (MPO), highly expressed in neutrophils, functions beyond antimicrobial activity and can modulate redox signaling that influences chromatin accessibility and metabolism in nearby immune and epithelial cells [24]. MPO-dependent oxidative signaling has been implicated in modifying chromatin accessibility in neighboring macrophages and epithelial cells, suggesting that neutrophil-derived reactive species may propagate trained or tolerant programs in surrounding mucosal populations [7,24,54]. Whether these reprogramming events confer truly long-lived “trained” states in neutrophils remains an area of investigation, yet evidence supports transient or progenitor-based memory that can persist through multiple neutrophil generations in chronic inflammatory contexts such as IBD [12,17,20,50]. Mounting evidence indicates that at least short-term memory-like phenotypes can exist in circulating or newly recruited neutrophils [47,48,52].

Furthermore, in neutrophils TLRs (e.g., TLR4 for LPS) and other receptors (e.g., dectin-1 for β-glucans) can become primed by initial encounters, leading to an amplified burst of reactive oxygen species, NET formation, or cytokine release upon re-stimulation. For example, TLR4 recognizes LPS from Gram-negative bacteria, while dectin-1 binds fungal β-glucans, each initiating distinct yet overlapping signaling cascades that can prime neutrophil functions [12,48,55]. Repeated or intense stimulation via these receptors can lead to heightened neutrophil responses, including excessive ROS, NETosis, or the secretion of inflammatory mediators [47,48,52]. When unchecked, these amplified responses contribute to epithelial injury and barrier dysfunction, highlighting the fine balance between protective and pathological neutrophil training [20,24]. Such priming can be further modulated by cytokines from macrophages or ILCs, as neutrophils rely heavily on paracrine signals in tissues. The interplay of multiple PRRs and cytokine-induced secondary signals supports a scenario in which neutrophils, too, can express memory-like reactivity—particularly in the context of repeated infections or chronic inflammatory settings [20,47].

Collectively, these findings support a model of short-term or progenitor-imprinted neutrophil memory that enhances host defense but may exacerbate chronic mucosal inflammation when persistently engaged.

### 3.2. Monocytes and Macrophages in Trained Immunity

Monocytes and macrophages were the first innate immune cell types shown to develop trained immunity, representing the most well-characterized model of this phenomenon. Classical training stimuli—such as β-glucan, Bacille Calmette–Guérin (BCG), and oxidized low-density lipoprotein (oxLDL)—induce long-term epigenetic and metabolic reprogramming that enhances proinflammatory responses upon restimulation [2,5,12,55,56]. Exposure to these stimuli leads to chromatin remodeling at key promoters including TNFA, IL6, IL1B, and PTGS2, resulting in sustained histone modifications (notably H3K4me3 enrichment and H3K9me3 depletion) and increased accessibility of enhancer elements linked to cytokine and metabolic genes [2,5,16,36].

Mechanistically, this reprogramming is driven by activation of the Dectin-1–Syk–CARD9 signaling axis and the downstream mTOR–HIF1α pathway, which couples immune signaling with glycolytic flux and cholesterol biosynthesis [6,57]. These metabolic alterations generate intermediates—such as fumarate, succinate, and acetyl-CoA—that act as cofactors for histone-modifying enzymes, reinforcing chromatin accessibility and maintaining a “poised” transcriptional state even after the initial stimulus is cleared [6,12,57].

In vitro and in vivo models have confirmed that β-glucan-trained monocytes exhibit increased glycolysis, mevalonate pathway activation, and glutaminolysis, collectively enabling rapid ATP production and cytokine synthesis during secondary challenges [6,58]. Conversely, certain stimuli such as high-dose LPS or chronic IFN_α/β_ exposure drive a tolerant phenotype characterized by H3K9me3 enrichment, decreased glycolysis, and increased oxidative phosphorylation [15,16]. Thus, macrophages and monocytes display a bidirectional capacity for training or tolerance, dictated by the intensity, duration, and metabolic context of the initial exposure.

In the intestinal mucosa, monocyte-derived macrophages undergo continuous education by the microbiota and local metabolites. SCFAs such as butyrate and propionate promote a tolerogenic phenotype by inhibiting HDACs and enhancing H3K27ac at genes associated with IL-10 production and tissue repair [40,46,59]. In contrast, microbial β-glucans and peptidoglycans drive proinflammatory training through Dectin-1 and NOD2 signaling, respectively, which converge on NF-κB and mTOR activation [55,57]. These opposing pathways exemplify how the mucosal environment determines whether trained immunity enhances protection or perpetuates inflammation.

Tissue-resident macrophages in the gut, derived from both embryonic precursors and monocyte influx, retain distinct “epigenetic memories” reflective of prior microbial exposure. Single-cell ATAC-seq studies of intestinal macrophages have revealed that commensal colonization induces persistent chromatin accessibility at metabolic and cytokine loci (e.g., Nos2, Slc2a1, Il10), suggesting that mucosal macrophages undergo continual reprogramming in response to microbial flux [36,60]. This plasticity enables rapid adaptation to luminal perturbations but also represents a vulnerability in chronic inflammatory states such as IBD, where repetitive microbial stimulation sustains pathogenic “trained” signatures.

Beyond the intestine, trained macrophages have been implicated in systemic inflammatory conditioning. BCG vaccination induces long-lived training in bone marrow progenitors, giving rise to circulating monocytes with heightened responsiveness for months to years [12]. Similarly, chronic Western-type diet exposure leads to “innate immune memory” within myeloid precursors through NLRP3 and IL-1β-dependent mechanisms, priming macrophages for exaggerated cytokine release during subsequent challenges [36,50,61]. These findings extend the concept of trained immunity beyond mature cells to encompass progenitor-level “central training,” whereby bone marrow hematopoietic stem and progenitor cells (HSPCs) acquire altered transcriptional states that propagate to their progeny [53,62].

Recent multi-omics analyses have uncovered distinct macrophage training signatures in GI disease contexts. In IBD, lamina propria macrophages display persistent H3K27ac and ATAC peaks near IL1B and CXCL8, consistent with trained inflammatory programming [36,63]. Conversely, probiotic or SCFA-enriched environments reverse these marks, restoring tolerogenic chromatin profiles and reducing proinflammatory cytokine output [40,64,65]. Collectively, these data demonstrate that macrophage training in the GI tract is both stimulus-specific and reversible, highlighting therapeutic potential for reprogramming maladaptive innate memory.

Functionally, trained macrophages in mucosal tissues enhance pathogen clearance and tissue repair by accelerating cytokine production, phagocytosis, and antigen presentation. Yet excessive or chronic training sustains a low-grade inflammatory state, contributing to epithelial damage and fibrosis. The duality of trained macrophage function—protective in acute infection but pathogenic in chronic disease—represents a central paradox of innate immune memory in the GI environment [1,58].

### 3.3. Dendritic Cells (DCs) and Trained Antigen Presentation

Dendritic cells act as sentinels that bridge innate and adaptive immunity. Beyond their classical antigen-presenting role, recent evidence shows that DCs can themselves acquire trained immunity, retaining altered transcriptional and metabolic profiles that influence subsequent T-cell activation [10,13,36,66]. Exposure to BCG, β-glucan, or commensal-derived peptidoglycans induces persistent changes in histone modifications such as H3K4me3 and H3K27ac at promoters for IL12, IL23, CD80, and CD86, leading to amplified IL-12p70 and IL-23 secretion upon restimulation [57].

This remodeling is reinforced by metabolic rewiring: trained DCs shift toward glycolysis and mevalonate pathway activation via mTOR-HIF1α signaling, paralleling the pathways described in trained monocytes [57]. Conversely, vitamin A-derived retinoic acid and epithelial TSLP can counterbalance these programs by promoting tolerogenic DCs that favor regulatory T cell (Treg) induction [60].

At mucosal surfaces, particularly in the intestine, DC training is shaped by constant microbial sampling. Clostridial SCFAs limit DC inflammatory potential by inhibiting HDACs, whereas Enterobacteriaceae LPS primes DCs for IL-12-rich Th1 responses [40,67]. DCs in Peyer’s patches exhibit “tissue memory,” maintaining open chromatin at metabolic and cytokine loci after transient inflammatory exposures [36,60]. This enduring flexibility enables rapid re-activation during secondary infection but can predispose to dysregulated inflammation if microbial composition changes abruptly.

Emerging single-cell ATAC-seq and CUT&Tag studies demonstrate that intestinal DC subsets retain distinct “training signatures,” with CD103^+^ DCs biased toward tolerogenic reprogramming and CD11b^+^ DCs biased toward inflammatory training [36,63]. Collectively, these findings place DCs as key orchestrators of both adaptive polarization and innate memory within mucosal niches.

### 3.4. Natural Killer (NK) Cells and Cytokine-Driven Innate Memory

Natural killer cells exhibit a unique form of antigen-independent memory, known as cytokine-induced or adaptive-like NK memory. Following exposure to viral antigens (e.g., CMV) or cytokines such as IL-12, IL-15, and IL-18, NK cells undergo stable epigenetic reprogramming that enhances their effector functions for weeks [13,62,66,68]. These NK memory-like cells show persistent chromatin accessibility at IFNG, PRF1, and GZMB promoters, accompanied by increased histone acetylation and decreased DNA methylation [62].

Metabolically, trained NK cells exhibit augmented glycolytic capacity and mitochondrial biogenesis through sustained mTORC1 and c-Myc activation, supporting rapid cytokine release and cytotoxic granule loading [68]. BCG vaccination, β-glucan priming, and even certain gut-derived metabolites such as inosine and butyrate have been reported to modulate NK training [13,57]. In murine models, BCG-induced NK training enhances clearance of Listeria and Candida infections via IFN-γ-dependent pathways, providing direct in vivo evidence of durable NK memory.

In the intestinal environment, macrophage- or DC-derived IL-18 and IL-12 promote local NK training, enabling heightened surveillance of viral or transformed epithelial cells [22,63]. However, persistent IL-15 signaling, common in chronic inflammation or IBD, can drive NK hyper-activation and tissue injury [36,63]. Thus, NK training represents a double-edged sword—bolstering mucosal defense yet amplifying pathology when regulation fails.

### 3.5. Innate Lymphoid Cells (ILCs) and Tissue-Adapted Training

ILCs are increasingly recognized as innate counterparts to T cells, capable of mounting rapid cytokine responses upon repeated stimulation. Recent work demonstrates that ILC subsets undergo training-like reprogramming shaped by cytokine milieu and microbial metabolites [57,59,63,69].

ILC1 plasticity: ILC1s, exposed to IL-12 and IL-18, develop long-lived IFN-γ-competent states reminiscent of NK memory cells. This adaptive-like transition supports antiviral defense yet can drive epithelial apoptosis in persistent infection models [66,68,69].

ILC2 training: Repeated IL-33 or TSLP exposure induces stable H3K27ac enrichment at Il5 and Il13 enhancers, leading to heightened type-2 cytokine release during subsequent challenges. Metabolic analyses reveal increased fatty-acid oxidation and glycolytic plasticity in trained ILC2s, supporting prolonged cytokine secretion and survival. In allergic inflammation, this persistent activation underlies chronic eosinophilia and fibrosis [56,57,70].

ILC3 training: IL-1β and aryl-hydrocarbon-receptor (AhR) ligands derived from dietary tryptophan metabolites reprogram ILC3s toward glycolytic metabolism and open chromatin at Il22 and Il17f loci [59,69]. These trained ILC3s enhance epithelial barrier repair and antimicrobial peptide expression but may perpetuate chronic colitis when overactivated [36,63].

Collectively, these findings reveal that mucosal ILCs integrate epithelial and microbial cues into heritable transcriptional states that dictate their long-term responsiveness. Single-cell multi-omic analyses in mouse and human gut now identify ILC-specific “trained” enhancers overlapping metabolic and cytokine pathways [41,63].

### 3.6. Integrative Perspective: Conserved Logic of Innate Training Across Cell Types

Across neutrophils, macrophages, NK cells, DCs, and ILCs, a common mechanistic framework emerges: transient stimulation induces metabolic rewiring (glycolysis, cholesterol synthesis, and fumarate accumulation) coupled with epigenetic remodeling (H3K4me3, H3K27ac, reduced H3K9me3) that renders cells or their progeny transcriptionally poised for rapid secondary responses [58,71,72]. Despite distinct ontogeny and lifespan, these lineages converge on similar signaling axes—mTOR-HIF1α, NOD2, and β-glucan-Syk-CARD9—that link nutrient sensing and chromatin accessibility.

A defining feature at mucosal surfaces is the integration of environmental metabolites—SCFAs, indole derivatives, and bile acids—which determine whether the trained state is inflammatory or tolerogenic [40,43]. This concept of “contextual training” underscores that the same pathway can yield protection or pathology depending on microbial composition and cytokine milieu.

In homeostasis, trained mucosal immunity enhances resistance to reinfection and accelerates barrier repair. However, chronic activation or dysbiosis-driven retraining can create maladaptive loops that sustain inflammation in diseases such as IBD, IBS, and colorectal cancer [63,73,74,75]. Systems-level studies integrating scRNA-seq, ATAC-seq, and metabolomics are now elucidating lineage-specific and shared signatures of training, opening new opportunities for therapeutic modulation of innate memory [29,36,60,63]. A comparative summary of the shared and cell-specific mechanisms of innate immune training is presented in Table 1, highlighting conserved metabolic–epigenetic axes and unique features across innate immune lineages.

## 4. Overview of Mucosal Innate Immunity in the GI Tract

The GI mucosa comprises a single layer of epithelial cells covered by mucus, which together form a physical and biochemical barrier [8]. This barrier must simultaneously allow nutrient absorption and symbiotic microbial interactions while preventing pathogen invasion. The intestinal epithelium undergoes rapid turnover, a process crucial for tissue homeostasis and barrier maintenance [3,8,22,27]. For example, epithelial tight junctions regulate selective paracellular permeability in a surprisingly dynamic fashion. While epithelial dynamics are central to mucosal physiology, the primary focus of mucosal trained immunity lies within the immune cell compartment—macrophages, dendritic cells, neutrophils, and innate lymphoid cells (ILCs)—that constantly interact with epithelial and microbial cues to sustain balance.

Intestinal epithelial cells consist of a diverse group of specialized cells including Paneth cells, tuft cells, goblet cells, enteroendocrine cells, microfold (M) cells, and enterocytes, many of which have critical innate immune functions. For instance, Paneth cells produce antimicrobial peptides in the small intestine [84]. Other mucosal cells such as enterocytes produce RegIIIγ, an antimicrobial peptide [84]. Goblet cells throughout the intestinal tract secrete mucus that helps limit bacterial adhesion [84]. Tuft cells are known for their roles in anti-helminthic and antimicrobial defense through the secretion of IL-25, recruitment of innate lymphoid cells (ILCs), sensing microbial populations shifts through detection of the short chain fatty acid succinate, and secretion of proinflammatory mediators such as leukotrienes and prostaglandins [80,82]. Additionally, M cells sample luminal contents and facilitate the transcytosis of luminal antigens across the epithelial barrier to other immune cells in gut associated lymphoid tissue such as Peyer’s patches or isolated lymphoid follicles [84]. These epithelial–immune interactions provide the contextual cues that instruct innate immune training or tolerance. For instance, IL-33, thymic stromal lymphopoietin (TSLP), and IL-25 released by epithelial cells shape the responsiveness of ILC2s and macrophages, integrating barrier status with immune readiness [57,80,82].

Several innate cell populations coordinate immune defense and homeostasis in the gut. Macrophages are abundant in the lamina propria, playing essential roles in phagocytosis, cytokine production, and tissue remodeling, typically exhibiting a tolerogenic phenotype in healthy intestines [85,86]. DCs in Peyer’s patches and the lamina propria sample luminal antigens and orchestrate tolerance or adaptive responses in their interactions with local T cell populations [37,81,85,87]. ILCs (subsets ILC1, ILC2, and ILC3) modulate barrier function, inflammation, and tissue repair, responding rapidly to cytokines and microbial signals [21,87]. Neutrophils are first responders in acute infection or inflammation^43^, and epithelial cells, though not traditionally classified as immune cells, express pattern recognition receptors (PRRs) that enable them to detect microbial patterns and secrete immunomodulatory factors, as described above [26]. In this regard, Th17-derived IL-17 acts on stromal cells to recruit neutrophils and promote a neutrophil-supportive environment within the intestine [10,63]. Eosinophils, mast cells, and basophils in the gut play important roles in anti-helminthic immunity and allergy. Eosinophils in particular have also been reported to maintain small intestinal villous homeostasis, modulate barrier integrity, and help shape macrophage differentiation [88]. They have also been shown to promote Th2 responses and suppress Th1 responses in the intestine [89,90]. Similarly, mast cells, specifically mast cell chymase, have been shown to regulate barrier integrity in the small intestine [91]. These populations constitute a highly interconnected mucosal network in which epithelial signals calibrate the training or tolerance of innate immune cells, ensuring barrier repair, microbial containment, and controlled inflammation [36,60,63]. Together, the intestinal innate immune system maintains an important immunological and physical barrier critical to epithelial homeostasis.

## 5. Specifics of Innate Immune Memory in the Gastrointestinal Tract

The GI tract is a key component of the mucosal immune system. Many cell types (as previously discussed) contribute to the innate immune system within the GI tract. While many innate immune cells reside in the GI tract during homeostasis, and these innate immune cells participate in trained immunity elsewhere in the body, the role of trained immunity in the gut for every innate immune cell type is not definitively established. Many intestinal studies have focused on innate immune training of macrophages, monocytes, neutrophils, and ILCs. The intestinal microbiota and products of their metabolism serve as the main inducers of innate immune memory in the GI tract.

The gut microbiota includes bacteria, archaea, viruses, and fungi that collectively outnumber the host’s somatic cells [8]. Microbial-derived metabolites (e.g., SCFAs such as butyrate) promote regulatory pathways that maintain tolerance, while dysbiosis (an imbalance in microbial composition) is linked to chronic inflammatory conditions like inflammatory bowel disease (IBD) and metabolic syndrome [74,92]. Microbial signals also tune resident macrophages, dendritic cells (DCs), and ILCs toward either tolerogenic or inflammatory phenotypes, depending on context [21,93].

The unique characteristics of the GI mucosa—constant microbial exposure, rapid epithelial turnover, and continuous antigen sampling—make it an ideal system for studying both trained and tolerogenic innate immunity. Repeated microbial or dietary stimuli drive continuous remodeling of mucosal macrophage, DC, and ILC transcriptional networks. This dynamic state supports “context-dependent training,” where identical pathways (e.g., NOD2, TLR4, or Dectin-1) can lead to either hyperresponsiveness or tolerance, depending on local metabolite and cytokine composition [21,49,93].

### 5.1. Microbiota Induced Trained Immunity

Both commensal and pathogenic microbiota have been known to induce trained memory phenotypes in innate immune cells. Commensal microbes are microbes which can co-exist with minimal negative effects on the host. For example, the commensal and probiotic bacteria *Lactiplantibacillus plantarum* has been shown to modulate human monocyte and macrophage responses after bacterial contact [94]. *L. plantarum* can reside intracellularly in macrophages, and previous macrophage contact with this microbe improves bacterial intracellular survival and decreases macrophage secretion of TNFα [94]. *L. plantarum* exposure modulated RNA transcriptional profiles of macrophages, especially in histone associated genes [94]. Specifically, pre-exposure of *L. plantarum* to macrophages decreased their reactive oxygen species production, facilitating improved bacterial survival [94].

Another commensal microbe, *Akkermansia muciniphilia* promotes a similar response; *A. muciniphilia* can also survive intracellularly in human and murine macrophages [95]. Training of murine and human macrophages with either heat inactivated *A. muciniphilia,* then live *A. muciniphilia* or repeated stimulation with *A. muciniphilia* alone decreased macrophage secretion of IL-6 and TNFα and downregulated NF-κB [95]. *A. muciniphilia* trained macrophages exhibited an increase in glycolysis and mitochondrial respiration [95]. These data highlight a role for bacterial hijacking of innate immune memory to promote their own survival.

Other commensals, such as *Bacteroides fragilis* and *Faecalibacterium prausnitzii*, imprint tolerogenic macrophage and DC programs through the production of polysaccharide A or butyrate, respectively, promoting IL-10 and TGF-β secretion and restraining inflammatory training [36,40,43,61]. Conversely, pathobionts including *Enterococcus faecalis* and adherent-invasive *E. coli* can induce inflammatory trained states via sustained NOD2 and TLR5 activation, fueling chronic colitis [36,60].

At the progenitor level, chronic microbial stimulation has been shown to imprint hematopoietic stem cells (HSCs) in the bone marrow, biasing myelopoiesis toward a “trained” proinflammatory phenotype detectable in circulating monocytes [11,17,18]. This axis—gut inflammation → microbial metabolites → bone marrow reprogramming—illustrates how mucosal innate memory can exert systemic effects.

### 5.2. Short Chain Fatty Acids

SCFAs, including acetate, propionate, and butyrate, are the major products of dietary fiber metabolism by gut commensal bacteria. Beyond their role as energy substrates for colonocytes, SCFAs act as potent immunomodulatory metabolites that shape trained immunity through both epigenetic reprogramming and metabolic rewiring [40,43].

Butyrate and propionate, for example, condition monocytes toward tolerogenic programs by reducing IL-6 and TNF secretion upon bacterial or TLR stimulation [43]. In colonic macrophages, butyrate suppresses nitric oxide, IL-6, and IL-12 production, an effect mediated through inhibition of HDACs and increased H3K9 acetylation [40]. Importantly, these effects are independent of canonical SCFA receptors such as GPR41 and GPR43, emphasizing a direct epigenetic mechanism [40,43].

SCFAs also influence other innate and mucosal immune populations. Dendritic cells exposed to SCFAs reduce their antigen presentation capacity and IL-12 secretion, creating conditions favorable for regulatory T cell induction [46]. In neutrophils, acetate has been reported to regulate chemotaxis and oxidative burst activity, while butyrate dampens NETosis and excessive ROS production [96]. Innate lymphoid cells (particularly ILC3s) sense SCFA availability to fine-tune IL-22 production, linking microbiota-derived metabolites directly to barrier repair and antimicrobial defense [67].

SCFAs further protect epithelial barrier integrity by enhancing tight junction protein expression, stimulating mucus production, and serving as the primary fuel for colonocytes [65,97,98]. These effects not only preserve barrier function but also reduce translocation of luminal antigens that could otherwise perpetuate inflammatory training of macrophages and dendritic cells.

Recent studies extend these findings to cross-kingdom signaling: butyrate and propionate influence mitochondrial function and chromatin openness in epithelial cells, indirectly regulating macrophage and DC training through epithelial-derived mediators such as IL-18 and TSLP [36,60]. SCFA insufficiency, common in fiber-poor diets or dysbiotic states, shifts the intestinal milieu toward proinflammatory macrophage and neutrophil phenotypes [40,43].

Clinically, a reduction in SCFA output, often tied to depletion of fiber-fermenting taxa like *Faecalibacterium prausnitzii* or *Roseburia*, has been associated with both IBD and IBS [74]. Insufficient SCFAs appear to shift mucosal immunity toward a proinflammatory bias, enhancing activation of macrophages and neutrophils. Conversely, therapeutic strategies such as high-fiber diets, SCFA supplementation, or prebiotic/probiotic interventions are being explored to restore a tolerogenic trained immunity profile in the gut [65,98].

Other microbial metabolites—including indole derivatives (via tryptophan catabolism) and secondary bile acids—also act as “metabolic trainers.” Indole-3-propionate and lithocholic acid promote IL-10 expression in macrophages via AhR and FXR activation, reinforcing anti-inflammatory training [79,83,99]. Conversely, excessive bile acid deconjugation by dysbiotic microbiota increases proinflammatory macrophage polarization through TGR5 inhibition [41,100]. These parallel metabolite circuits illustrate that SCFAs are part of a broader chemical network that orchestrates mucosal trained immunity.

These findings illustrate a broader concept: microbial metabolites can serve as key regulators of immune tone in the intestine. By influencing both cellular metabolism and chromatin accessibility, SCFAs adjust innate cell function in ways that enhance antimicrobial protection while limiting unnecessary inflammation.

Together, microbiota-induced training and metabolite-mediated reprogramming create a layered regulatory architecture in the GI tract. This architecture integrates microbial ecology, nutrient availability, and immune cell epigenetics to maintain equilibrium. Disruption of this balance—through antibiotics, low-fiber diets, or chronic inflammation—leads to maladaptive training that fuels disease persistence. Understanding and therapeutically redirecting these pathways offers promising strategies for restoring immune homeostasis in mucosal disease [63].

## 6. Innate Immune Memory and GI-Related Diseases

### 6.1. Inflammatory Bowel Diseases: Crohn’s Disease and Ulcerative Colitis

Crohn’s disease (CD) and ulcerative colitis (UC) are classical examples of chronic mucosal inflammation in which innate immune dysregulation plays a central role. Evidence increasingly indicates that maladaptive trained immunity as a driver of their persistence and the relapsing nature [68,101].

Intestinal macrophages and monocytes in IBD patients often show epigenetic and metabolic signatures consistent with training, including increased H3K4me3 enrichment at proinflammatory loci and augmented glycolytic flux. These features correlate with exaggerated secretion of TNF-α, IL-1β, and IL-6 upon microbial stimulation, fueling ongoing tissue damage [7,101]. Neutrophils recruited into inflamed mucosa can also adopt short-lived trained phenotypes, marked by increased oxidative burst and NET formation, both of which contribute to epithelial barrier injury and further immune activation [20,102].

Epithelial and stromal compartments also intersect with trained immunity programs. One consequence of barrier breakdown in IBD is increased passage of luminal microbes, which repeatedly engage pattern-recognition receptors on innate immune cells [10,26,37,103]. This ongoing microbial stimulation promotes the persistence of trained programs in macrophages and dendritic cells, amplifying mucosal inflammation in a feed-forward manner. Moreover, trained immunity can interact with adaptive immunity by skewing dendritic cells and ILCs toward proinflammatory phenotypes that enhance Th1/Th17 responses, both of which are pathogenic in IBD [18,21].

Recent single-cell and epigenomic analyses have shown that lamina propria macrophages from active IBD lesions maintain open chromatin at IL1B, CXCL8, and S100A8/A9, reflecting “locked-in” inflammatory training [29,66,71]. Similarly, intestinal fibroblasts exhibit trained-like transcriptional memory following TNF and IL-1β exposure, maintaining a proinflammatory transcriptome that drives matrix remodeling and fibrosis [29,66,71]. These data extend the concept of innate memory beyond immune cells to stromal components of the gut wall.

Chronic stimulatory microbiota-derived signals are central to this process. Dysbiosis in IBD is characterized by a loss of butyrate-producing bacteria and enrichment of pathobionts such as adherent-invasive *Escherichia coli* [40,43,74]. Reduced butyrate weakens tolerogenic training of colonic macrophages, while persistent exposure to microbial flagellin, LPS, or fungal β-glucans promotes maladaptive proinflammatory reprogramming [10,16,85]. In this sense, the quality of training stimuli—commensal versus pathogenic—determines whether trained immunity supports mucosal homeostasis or drives chronic inflammation, as summarized in Figure 2.

Mechanistically, maladaptive trained immunity in IBD involves chronic activation of NOD2, TLR4, and IL-1R signaling, which continuously fuels the mTOR–HIF1α metabolic axis and sustains glycolysis and cholesterol biosynthesis [6,12,26,37,38]. Parallel mitochondrial dysfunction increases succinate accumulation, which stabilizes HIF1α and drives IL-1β expression—an amplifying metabolic loop [45]. Together, these features constitute a self-perpetuating inflammatory memory circuit.

Therapeutic strategies are beginning to leverage this knowledge. Epigenetic modifiers, such as histone deacetylase inhibitors, have shown efficacy in preclinical models by dampening hyper-responsive myeloid cells [31,66]. Agents targeting mTOR (e.g., rapamycin analogs) or fumarate metabolism (e.g., dimethyl fumarate) have also demonstrated capacity to “reset” macrophage training and restore homeostatic metabolism [6]. Microbiome-targeted therapies, including specific probiotics and fecal microbiota transplantation (FMT), may help reintroduce tolerogenic microbial signals that “reset” trained immunity programs [64,74]. The development of trained immunity–based vaccines, such as MV130, represents another promising avenue, as such approaches can bolster mucosal defense while curbing uncontrolled inflammation [104]. Importantly, genome-wide association studies linking UC and CD susceptibility loci to genes involved in trained immunity underscore the potential clinical relevance of these pathways [74].

Taken together, dysregulated trained immunity provides a unifying framework to understand how environmental triggers (microbiota, diet, infections) and host genetics converge to sustain chronic inflammation in IBD. Future interventions that selectively reprogram maladaptive trained states—while preserving protective innate memory—represent an emerging therapeutic frontier [18,63].

### 6.2. Irritable Bowel Syndrome (IBS) and Disorders of Gut–Brain Interactions (DGBI)

IBS is among one subtype of DGBIs, formerly termed “functional” GI disorders, defined primarily by altered bowel habits and visceral hypersensitivity. Yet increasing evidence suggests that immune dysregulation and barrier defects contribute meaningfully to its pathogenesis.

Several studies have documented low-grade mucosal immune activation in subsets of IBS patients, including increased numbers of activated mast cells, eosinophils, and macrophages in the lamina propria [73,105,106,107]. These immune populations release mediators such as histamine, tryptase, and cytokines that sensitize enteric nerves and disrupt epithelial tight junctions, thereby amplifying visceral hypersensitivity and motility disturbances [73,108].

Trained immunity may provide a mechanistic explanation for the persistence of immune activation in IBS. Chronic exposure to dysbiotic microbial signals or subclinical infections can induce epigenetic and metabolic reprogramming in intestinal macrophages, mast cells, or ILCs, resulting in heightened responsiveness upon repeated stimulation. For example, monocytes trained by bacterial components or microbial metabolites may maintain elevated production of IL-6, IL-1β, or TNF, which in turn reinforces epithelial barrier alterations and neuronal sensitization [60,79,83,105]. This low-grade but persistent inflammatory tone aligns with the clinical observation that symptoms often wax and wane but rarely resolve completely [73].

Recent metabolomic and scRNA-seq studies in IBS colonic biopsies reveal increased lactate and succinate accumulation and upregulation of glycolytic enzymes in mucosal macrophages, suggestive of low-level mTOR–HIF1α activation consistent with trained metabolic profiles [45,60]. Additionally, epigenetic profiling of IBS macrophages has identified enhanced H3K4me3 and reduced H3K9me3 at inflammatory gene promoters, paralleling trained immunity signatures observed in mild IBD [7,41].

Microbial metabolites are central to this process. Butyrate and related SCFAs normally dampen inflammation and promote tolerance, but reduced abundance of butyrate-producing taxa in IBS may deprive the mucosa of this protective influence. Altered bile acid metabolism and excess tryptophan catabolites further contribute to immune memory shifts that favor inflammation [65,98,109,110]. Indole derivatives (e.g., indole-3-aldehyde) and serotonin metabolites modulate macrophage and ILC3 function via AhR and 5-HT receptors, respectively, influencing tolerance and epithelial repair [79,83]. Perturbations in these pathways likely contribute to the persistence of low-grade immune activation and dysmotility in IBS.

Recognizing IBS as a disorder influenced by mucosal trained immunity has practical value. It provides a framework to understand why certain patients benefit from probiotics, dietary fiber, or fecal microbiota transplantation—interventions that may shift metabolite availability and recalibrate immune memory [109]. Additionally, epigenetic-targeting therapies currently explored in IBD, such as histone deacetylase inhibitors, may have relevance for IBS if safety and dosing can be optimized. Importantly, trained immunity may also help account for the long-term symptom persistence observed in post-infectious IBS, where a single infectious insult imprints a heightened immune responsiveness that continues to influence mucosal physiology months to years later [110].

Future research using integrated scATAC-seq and metabolomics approaches could identify IBS-specific “trained signatures” distinguishing subtypes (diarrhea-predominant vs. constipation-predominant vs. mixed), paving the way for metabolically guided precision therapies [7,36,63,79,83].

### 6.3. Infections and Mucosal Pathogens

Recurrent infections such as *Clostridioides difficile*, *Salmonella*, or *Helicobacter pylori* can induce innate immune training, potentially leading to improved pathogen clearance upon re-infection [111,112]. However, excessive or repeated stimulation can also promote a chronic inflammatory state, heightening the risk of tissue damage and post-infectious complications [22,73,112].

Murine exposure to a gut-evolved strain of *Candida albicans* protected mice from mortality when subsequently exposed to a fully virulent strain of *C. albicans* [53]. This training was found to be independent of typical receptors involved in trained immunity such as Dectin-1, NOD2, TLR4, and TLR2 [48,53,95]. Instead, this phenotype is dependent on neutrophils, and caused an increase in glycolytic activity and mitochondrial respiration, as seen in other trained immunity studies [95,113]. *Citrobacter rodentium* infection promotes mucosal ILC3 expansion and promoted survival upon reinfection [113]. ILC3 trained immunity was dependent on IL-22 production, and these ILC3s exhibited marked metabolic shifts to oxidative phosphorylation and fatty acid oxidation, from glycolysis. Of note, ILC3 training also occurred after repeated *Listeria monocytogenes* infection, suggesting that ILC3 populations can be trained by multiple bacterial pathogens [59,69,113].

Multiple enteric viruses also induce trained immunity. For instance, Enterovirus 71 induced IL-25 dependent trained immunity in intestinal tuft cells [62]. Exposure to either IL-25 or Enterovirus 71 early in murine life induced greater tuft cell proliferation upon re-infection, and IL-25 training specifically reduced Enterovirus burden [62]. Understanding trained immunity mechanisms could inform prophylactic approaches or adjunct therapies that pre-empt hyperinflammatory responses in recurrent GI infections.

Additional evidence now implicates norovirus and rotavirus in shaping mucosal innate memory: both infections generate persistent epigenetic reprogramming of macrophages and ILC3s that enhances IL-22 and IFN-λ responses upon reinfection [10,62]. However, chronic or repeated viral exposure can exhaust these trained programs, transitioning from protection to tolerance.

Understanding the balance between beneficial and pathogenic training during infection may allow for novel vaccine strategies that exploit trained immunity. Live-attenuated or metabolic-adjuvanted mucosal vaccines (e.g., β-glucan–conjugated or SCFA-supplemented formulations) are being developed to promote protective but self-limiting innate memory [12,13,30,104].

## 7. Conclusions

Marked advances in our understanding of trained immunity have occurred since the emergence of this field. Innate immune cells including NK cells, macrophages, neutrophils, and ILCs exhibit trained immunity in the GI tract in various contexts [10,20,24,47,59,69,102] (Figure 3). Much of mucosal training is a result of microbiota, microbiota-derived components (LPS, flagellin, etc.), or microbiota-derived metabolites such as SCFAs [21,40,43,63,74]. Understanding molecular mechanisms behind trained immunity may provide novel therapeutic targets in the context of GI infections and inflammatory diseases [10,18,101]. Viewing IBD and IBS through the lens of mucosal trained immunity may help explain why some patients respond to probiotics, fiber supplementation, or fecal microbiota transfer, as these approaches could adjust the metabolite landscape and temper maladaptive immune memory [65,74,98,109,110].

The gut represents a unique site for the study of trained immunity, where continuous microbial exposure, epithelial turnover, and metabolic flux generate a dynamic immunologic landscape [8,22,27,74,80]. Unlike classical peripheral tissues, intestinal innate immune memory operates at the intersection of commensal tolerance and rapid pathogen defense [10,21,93]. This balance determines whether trained immunity supports mucosal homeostasis or fuels chronic inflammation [63,66,101].

Several important themes are emerging. First, it is becoming clear that the epigenetic and metabolic programs underlying trained immunity are not uniform but vary significantly depending on the stimulus, the tissue microenvironment, and the cellular lineage involved [2,5,10,54,114]. In the gut, this variability is magnified by the diversity of microbial cues and the constant renewal of the epithelial barrier [8,22,74,93]. These factors make the GI tract a uniquely dynamic model for dissecting how innate immune memory can be beneficial in pathogen defense while simultaneously predisposing to chronic inflammation when dysregulated [63,101].

There is growing recognition that trained immunity in mucosal tissues does not act in isolation. Crosstalk between innate and non-hematopoietic compartments—epithelial, stromal, and neural—shapes the quality and persistence of trained responses [57,80,82]. For example, epithelial sensing of SCFAs and tryptophan-derived indoles can modulate macrophage histone acetylation and control IL-10 versus IL-1β output, which in turn influences adaptive lymphocyte priming [40,43,79]. Stromal and fibroblast populations also exhibit “metabolic memory,” sustaining cytokine networks long after initial stimulation. Future research that integrates epithelial, stromal, and immune training mechanisms will provide a more complete picture of barrier immunity and how it adapts to chronic environmental change [7,36,63].

It is becoming apparent that trained immunity has a temporal dimension—short-term reprogramming in mature cells and long-term “central training” imprinted in bone marrow progenitors [12,20,106]. These layers may cooperate to sustain chronic inflammatory states or confer long-lasting protection depending on the context [18,45,101]. Understanding this hierarchy will be essential for therapeutic targeting [31,104].

Translation opportunities are beginning to take shape. Trained-immunity–based vaccines and microbiome-targeted therapies are now being tested in experimental colitis and recurrent infection models demonstrating the potential to reshape mucosal immune tone [12,13,30,64,104]. Epigenetic drugs, metabolic modulators (e.g., fumarate derivatives, mTOR inhibitors), and SCFA-mimetic compounds are being explored as “memory modifiers” to recalibrate hyper-responsive innate populations [31,40,43]. The challenge moving forward is to identify microbial or metabolic interventions that elicit protective training without tipping the balance toward pathology [63]. Advances in single-cell multiomics, chromatin profiling, and metabolic tracing will accelerate this discovery pipeline by mapping distinct training trajectories in human tissues [10,18,63].

Finally, the pleiotropic nature of trained immunity underscores its broad biomedical relevance. Links are emerging between mucosal trained immunity and systemic conditions such as metabolic syndrome, autoimmunity, neuroinflammation, and even cancer [18,36,63]. The gut, as the largest immune interface in the body, may act as a central “training hub” whose metabolic and microbial signals shape immune tone across distant organs [40,43,74]. Manipulating mucosal trained immunity could offer leverage points for restoring systemic homeostasis in diseases that extend well beyond the intestine [10,101].

The study of innate immune memory at mucosal sites has evolved from a conceptual novelty into a mechanistically grounded and translationally promising field [2,5,21,63]. By continuing to define the molecular circuits of training, distinguishing protective from pathogenic programs, and leveraging the microbiome as both an inducer and therapeutic target [40,43,64,65,74,98,104], we are poised to develop innovative strategies for modulating GI health and disease.

## Figures and Tables

**Figure 1 ijms-26-10093-f001:**
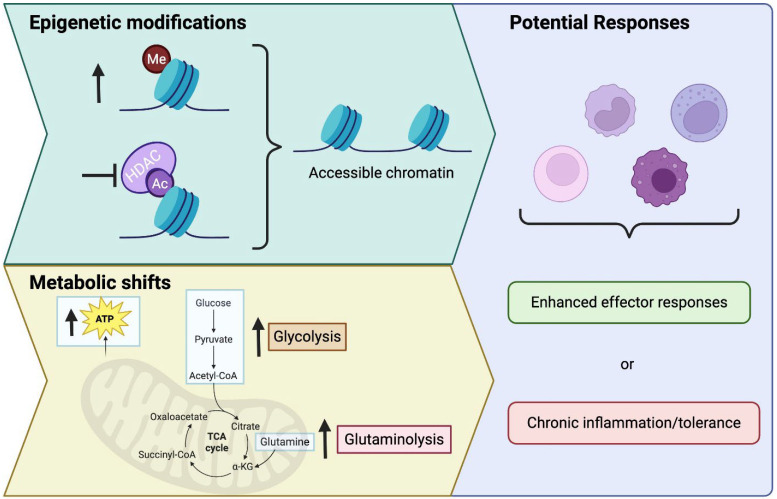
Overview of the mechanisms behind innate immune memory and potential responses. Epigenetic modifications such as increased histone methylation or decreased de-acetylation increase chromatin accessibility. Metabolic shifts may also occur, increasing glycolysis and glutaminolysis, thereby increasing ATP. These changes can lead to trained immunity responses such as enhanced effector responses, or in some cases, chronic inflammation and tolerance.

**Figure 2 ijms-26-10093-f002:**
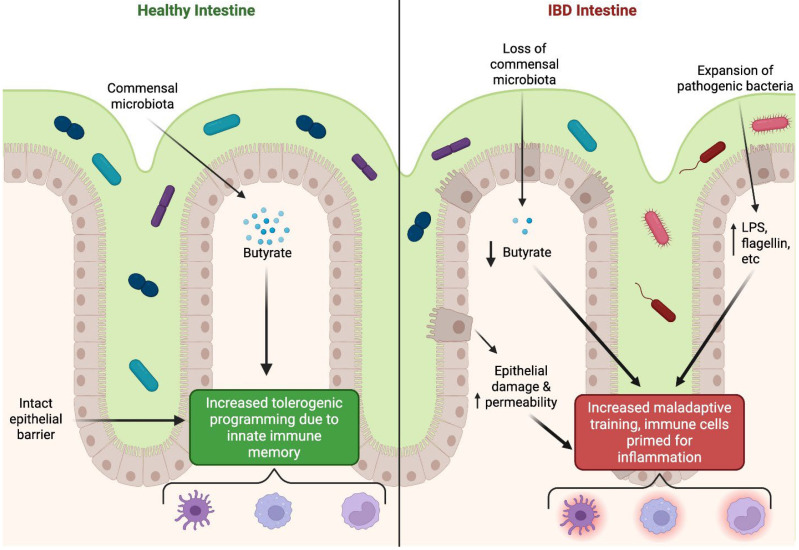
Interactions of trained immunity in IBD. In the healthy intestine commensal microbiota produce butyrate. This metabolite, coupled with an intact epithelial barrier that promote tolerogenic programing of innate immune cells such as DCs, monocytes, and macrophages. Conversely, in IBD there is a loss of commensal microbiota (teal) and an expansion of pathogenic bacteria (red) which leads to a decrease in butyrate production. In addition, epithelial damage increases permeability and thus allows for the passage of bacterial products (LPS, flagellin, etc) into the intestinal lamina propria where these signals prime DCs, monocytes, and macrophages for inflammation.

**Figure 3 ijms-26-10093-f003:**
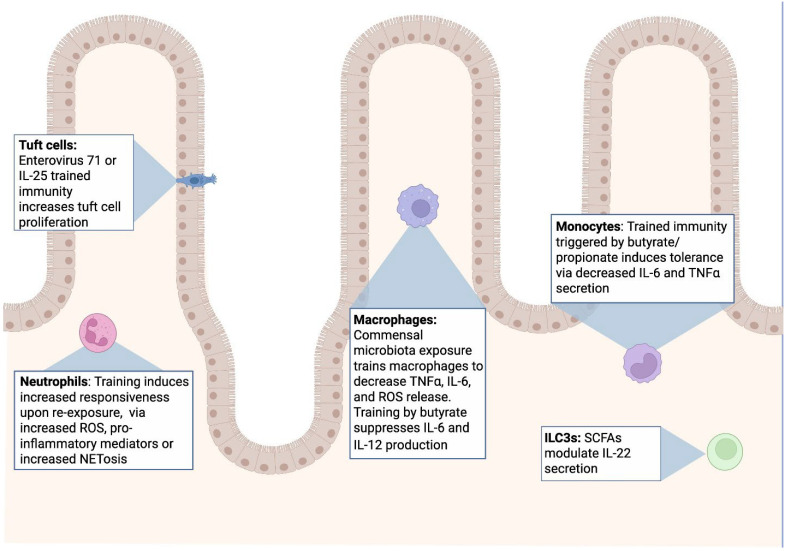
Examples of trained immunity within the GI tract.

**Table 1 ijms-26-10093-t001:** Shared and Cell-Specific Mechanisms Underlying Innate Immune Training.

Innate Cell Type	Shared Mechanistic Features	Distinct/Cell-Specific Mechanisms	Representative References
Common to all trained innate cells	Epigenetic remodeling: ↑ H3K4me3/H3K27ac; ↓ H3K9me3; enhanced chromatin accessibility at cytokine/metabolic loci; mTOR–HIF-1α axis linking glycolysis and lipid metabolism; accumulation of fumarate, succinate, acetyl-CoA reinforcing histone acetylation; interplay of metabolic intermediates and chromatin modifiers; balance between inflammatory “training” and tolerogenic “trained tolerance.”	—	[1,2,5,7,13,30,36]
Neutrophils	Share metabolic and redox-dependent pathways with monocytes.	Short-lived “trained granulopoiesis” (IL-1β/GM-CSF imprint HSPCs in bone marrow); MPO-dependent redox signaling modifies chromatin in neighboring cells; rapid glycolytic reprogramming; transient chromatin accessibility at oxidative/NETosis loci; reversible, progenitor-driven short-term memory (days).	[12,20,47,48,49,50,52,53]
Monocytes/Macrophages	Canonical model of trained immunity.	β-glucan/BCG activate Dectin-1–Syk–CARD9 → mTOR–HIF-1α; mevalonate and glutaminolysis pathways generate epigenetic cofactors; SCFAs induce tolerogenic “trained tolerance” via HDAC inhibition → ↑ IL-10, ↓ IL-6/TNF; distinct chromatin landscapes in intestinal macrophages shaped by microbiota.	[2,5,7,10,12,17,18,32,35,38,49,53,66,68,76,77]
Dendritic Cells	Share metabolic rewiring with monocytes.	β-glucan/peptidoglycan → persistent H3K4me3/H3K27ac at IL12, IL23, CD80/CD86; mTOR–HIF-1α-driven glycolysis/mevalonate metabolism; retinoic acid & TSLP induce tolerogenic DCs; CD103^+^ vs. CD11b^+^ DCs show distinct trained/tolerogenic chromatin states.	[10,36,57,60,63]
Natural Killer Cells	Epigenetic and metabolic circuits parallel macrophage training.	Cytokine-induced NK memory (IL-12/IL-15/IL-18) → stable accessibility at IFNG, PRF1, GZMB; mTORC1/c-Myc activation → glycolysis + mitochondrial biogenesis; trained by BCG, β-glucan, inosine, butyrate; persistent mucosal activation via IL-18/IL-12.	[13,62,66,68]
Innate Lymphoid Cells (ILCs)	Share epigenetic accessibility at cytokine and metabolic enhancers.	ILC2: repeated IL-33/TSLP → H3K27ac at IL5/IL13; ↑ FAO/glycolytic plasticity. ILC3: IL-1β/AhR ligands → H3K27ac at IL22/IL17F; switch to oxidative phosphorylation + FAO. ILC1: IL-12/IL-18 → long-lived IFN-γ-competent state; distinct tissue-adapted enhancer landscapes at mucosal surfaces.	[36,57,59,60,61,69,71,78]
Contextual modifiers (mucosal environment)	—	Microbial metabolites (SCFAs, indoles, bile acids) modulate trained vs. tolerant states through AHR, FXR, and TGR5; cytokine milieu (IL-10, IL-33, IL-25) and epithelial signals integrate metabolic context.	[57,62,79,80,81,82,83]

Table 1: Shared and cell-specific epigenetic and metabolic mechanisms underpinning innate immune training and tolerance across mucosal innate cell types. Abbreviations: FAO, fatty-acid oxidation; HSPC, hematopoietic stem and progenitor cell; MPO, myeloperoxidase.

## Data Availability

No new data were created or analyzed in this study. Data sharing is not applicable to this article.

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
