# Peer review of "Pleiotropic Mucosal Innate Immune Memory in the Gastrointestinal Tract"

_ijms, 2025, doi:10.3390/ijms262010093_

Round 1

Reviewer 1 Report

Comments and Suggestions for Authors

This review aims at covering the mechanisms underlying immune memory of innate immune cells in GI related diseases. The topic is still interesting and disease relevant even though several nice reviews in this field were published (PMID: 40279311,39794207, 38599162, 33293712, 35115374 etc). However, I have several major comments listed below"

  1. The review format is pretty disorganized, including reference, introduction and main sections.
  2. The authors did not cite proper reference when stating some conclusions, such as in the third paragraph of conclusion section. Authors need to comprehensively check if they give proper credits to original papers to support their perspective.
  3. The majority of cited reference is pretty old, even though the concept of innate memory was put forward 30 years ago. However, there are still many interesting findings published recently.
  4. The subsections did not flow really well. For instance, I anticipated the authors would specifically review some major innate immune cells after neutrophil's role, but they only summarized neutrophils. 
  5. The mechanisms for innate memory are very general and broad. The authors need to emphasize some similarity and difference utilized by different innate immune cells to maintain memory. 
  6. In section 'Overview of Mucosal Innate Immunity in the GI Tract', the authors introduced too much epithelial cells, which were not the key focus for this reveiw, even though broadly epithelial cells could be reckoned as innate immunity and also found to establish memory.
  7. Some typos are detected. The authors need to carefully check their spelling. 

Author Response

Comment 1: The review format is pretty disorganized, including reference, introduction and main sections

Author Response: We appreciate this feedback and have restructured the manuscript for clarity and flow. The revised outline now follows a standard progression:

  1. Introduction → 2) Mechanisms of Trained Immunity (epigenetic + metabolic) → 3) Emerging Roles by Cell Type (neutrophils; monocytes/macrophages; dendritic cells; NK cells; ILCs) → 4) Overview of Mucosal Innate Immunity in the GI Tract (brief epithelial context only) → 5) Specifics of GI Trained Immunity (microbiota, SCFAs, other metabolites) → 6) Disease Sections (IBD, IBS, infections) → 7) Therapeutic/Translational Outlook → 8) Conclusion.
    We also standardized reference style, eliminated duplicates, and aligned all in-text citations with the reference list numbering.

Comment 2: The authors did not cite proper reference when stating some conclusions, such as in the third paragraph of conclusion section. Authors need to comprehensively check if they give proper credits to original papers to support their perspective

Author Response: We conducted a line-by-line citation audit and inserted supporting references.

Comment 3: The majority of cited reference is pretty old… there are still many interesting findings published recently.

Author Response: We have substantially refreshed the bibliography (2023–2025) and integrated recent primary studies and state-of-the-art reviews, including (selection). These additions update sections on neutrophil training, ILC/NK plasticity, stromal memory, and translational therapeutics ). We retained seminal older papers where foundational.

Comment 4: The subsections did not flow really well. For instance, I anticipated the authors would specifically review some major innate immune cells after neutrophil’s role, but they only summarized neutrophils.

Author Response: We have expanded and re-ordered the cell-type sections to improve flow and completeness. Following neutrophils, we now include dedicated, balanced subsections on monocytes/macrophages, dendritic cells, NK cells, and ILCs , each with mechanisms, mucosal relevance, and recent citations (2023–2025). Cross-references guide the reader from mechanisms → cell types → GI disease contexts.

Comment 5: The mechanisms for innate memory are very general and broad. The authors need to emphasize some similarity and difference utilized by different innate immune cells to maintain memory.

Author Response:  We agree and have added a comparative ‘Shared vs Cell-Specific Mechanisms’ summary in the mechanisms section, plus a table (now Table 1) highlighting:

  • Shared features: H3K4me3/H3K27ac gain; reduced H3K9me3; mTOR–HIF-1α axis; glycolysis/mevalonate/fumarate loops.

  • Cell-specific features:

    • Neutrophils—short-lived reprogramming via trained granulopoiesis, MPO-redox signaling.

    • Monocytes/Macrophages—Dectin-1–Syk–CARD9 → mTOR/HIF-1α; mevalonate/glutaminolysis; reversible tolerance vs training.

    • DCs—training via glycolysis/mevalonate; counter-regulation by retinoic acid/TSLP promoting tolerance.

    • NK cells—IL-12/15/18-driven epigenetic memory; c-Myc/mTORC1 metabolic program.

    • ILCs—ILC2 (IL-33/TSLP), ILC3 (IL-1β/AhR ligands), ILC1 (IL-12/18) with distinct chromatin and metabolic biases.

Comment 6: In section ‘Overview of Mucosal Innate Immunity in the GI Tract’, the authors introduced too much epithelial cells, which were not the key focus for this review…

Author Response: We trimmed epithelial content to maintain focus on innate immune cells. The revised overview now provides only the minimal epithelial context necessary to frame immune training cues (e.g., IL-33, TSLP, IL-25; metabolite sensing). Additional epithelial details that are tangential were removed or condensed to a brief paragraph.

Comment 7: Some typos are detected. The authors need to carefully check their spelling.

Author Response: We performed a comprehensive edit of the manuscript. 

Comment 8: This review aims at covering… several nice reviews in this field were published (PMID: 40279311, 39794207, 38599162, 33293712, 35115374 etc).

Author Response: We thank the reviewer and now explicitly position our review relative to recent high-quality reviews by incorporating and citing comprehensive overviews. Our distinct contribution is a GI mucosa–focused synthesis that (i) contrasts training vs tolerance vs trained tolerance in mucosal contexts; (ii) integrates microbiota/metabolite circuits (SCFAs, indoles, bile acids) with cell-type–specific epigenetic and metabolic programs; (iii) adds 2023–2025 updates on neutrophil training, ILC/NK plasticity, and stromal memory; and (iv) articulates therapeutic implications (trained-immunity–based vaccines, metabolic/epigenetic modulators) specifically for IBD/IBS/infections.

Reviewer 2 Report

Comments and Suggestions for Authors

This review article is of generally high quality, with a focused theme and clear logic. It comprehensively covers the mechanisms of gastrointestinal mucosal innate immune memory (trained immunity), involved cell types, inducing factors (such as microbiota and metabolites), functional implications in diseases (IBD, IBS, infections), and therapeutic prospects. The content is cutting-edge and well-structured. However, as peer-review suggestions, specific revisions can be proposed from the following aspects to enhance its rigor, readability, and academic impact:

1、The boundary between "trained immunity" and "tolerance" needs further clarification. The article mentions that trained immunity can lead to either enhanced responses or tolerance, but the mechanistic distinction between the two remains somewhat ambiguous.Clearly specify which stimuli tend to induce trained enhancement ,such as β-glucan or BCGand which favor tolerance ; introduce "trained tolerance" as a separate concept and discuss it briefly to avoid reader confusion. 2、The section on neutrophil trained immunity is relatively weak. Although the article acknowledges the role of neutrophils in trained immunity, the mechanistic description is vague.Add discussion of their short lifespan and the resulting constraints on epigenetic plasticity; cite more recent studies (2023–2025) that support the possibility of "short-term memory" or functional reprogramming in neutrophils. 3、The link between IBS and trained immunity remains somewhat speculative. Attributing IBS symptoms to trained immunity is still an emerging hypothesis.
Explicitly state that "the mechanism is not yet fully established" or "requires further clinical validation"; supplement the discussion with whether existing single-cell sequencing or epigenomic data from IBS patients support this model.

Author Response

Reviewer Comment 1: The boundary between "trained immunity" and "tolerance" needs further clarification. The article mentions that trained immunity can lead to either enhanced responses or tolerance, but the mechanistic distinction between the two remains somewhat ambiguous. Clearly specify which stimuli tend to induce trained enhancement (such as β-glucan or BCG) and which favor tolerance; introduce "trained tolerance" as a separate concept and discuss it briefly to avoid reader confusion.

Author response: We thank the reviewer for this insightful suggestion. To improve clarity, we have expanded the discussion of “trained immunity” versus “tolerance” to emphasize their mechanistic and functional distinctions. Specifically, we now delineate which stimuli promote enhanced immune training (e.g., β-glucan, BCG, and oxidized LDL) versus those that induce tolerogenic programming (e.g., chronic LPS exposure, SCFAs, or IL-10–rich environments). We have also introduced the concept of “trained tolerance” as a distinct yet related phenomenon characterized by durable suppression of inflammatory responses through sustained histone methylation (H3K9me3 enrichment) and metabolic shifts toward oxidative phosphorylation. This clarification has been added to Section 3 (“Emerging Roles for Innate Immune Cells in Trained Immunity,”) and referenced again in the Specifics of Innate Immune Memory in the Gastrointestinal Tract section to provide continuity.

Comment 2: The section on neutrophil trained immunity is relatively weak. Although the article acknowledges the role of neutrophils in trained immunity, the mechanistic description is vague. Add discussion of their short lifespan and the resulting constraints on epigenetic plasticity; cite more recent studies (2023–2025) that support the possibility of "short-term memory" or functional reprogramming in neutrophils.

Author Response: We appreciate the reviewer’s constructive feedback. The section on neutrophil-trained immunity has been substantially expanded to include a discussion of the temporal limitations imposed by neutrophil lifespan and how progenitor-level reprogramming (“trained granulopoiesis”) overcomes these.

Reviewer comment 3: The link between IBS and trained immunity remains somewhat speculative. Explicitly state that the mechanism is not yet fully established or requires further clinical validation; supplement the discussion with whether existing single-cell sequencing or epigenomic data from IBS patients support this model.

Author Response: We appreciate this important point. In the revised manuscript, we explicitly acknowledge that the relationship between IBS and trained immunity remains hypothetical and unproven, and that further validation through longitudinal single-cell epigenomic and metabolomic studies is required to establish causality. This clarification appears at the beginning of the IBS section. We also supplemented this discussion with recent metabolomic and scRNA-seq data from IBS mucosa, which show glycolytic and histone-mark enrichment patterns consistent with but not yet definitive for trained immunity. These additions ensure a balanced interpretation consistent with the current state of the field.